

# Aluminum phosphate sludge as a phosphorus source for maize production under low soil phosphorus availability

Ayaobu Tolofari[1], Theresa Adesanya[2], Francis Zvomuya[2] and Qiuyan Yuan[1]

[1] Civil Engineering, University of Manitoba, Winnipeg, Manitoba, Canada
[2] Department of Soil Science, University of Manitoba, Winnipeg, Canada

## ABSTRACT

**Background:** With increasing food demand as a consequence of the growing world population, there is a corresponding demand for additional sources of phosphorus (P). Alum-phosphate (Al-P) sludge is a by-product of wastewater treatment and can be a good source of P. In this study, the response of maize (*Zea mays* L.) to Al-P sludge was tested. Maize was chosen as the test crop due to its prevalent use as human and animal food and as a source of biofuel. The objective of the study was to investigate Al-P sludge as a source of P compared to a commercial fertilizer (monoammonium phosphate, MAP).

**Methods:** A growth chamber assay was conducted over four cropping cycles (45 d each). The application rate was 9.7, 19.4, 29.1 and 38.8 mg P kg$^{-1}$ dry soil. Amendments were applied once at the start of the first cropping cycle. Plants were harvested after each cycle and pots were re-seeded. Dry matter yield (DMY), total P uptake, Al-P uptake, soil total P and Olsen-P concentrations, pH, and EC were measured.

**Results:** DMY was significantly greater in pots amended with Al-P sludge than in pots treated with MAP. There was a significant rate × cropping cycle interaction effect on DMY with the differences among rates in cycle 1 different from those in cycle 4. Phosphorus uptake depended on cropping cycle, P source and P application rate. With sludge uptake higher than MAP in all cycles, the highest P uptake was observed at the highest application rate except for cycle 2 where this was observed at the rate of 29.1 mg kg$^{-1}$. For MAP, phosphorus recovery efficiency (PRE) at the highest rate was significantly greater than that at the lowest rate whereas PRE in cycle 1 was significantly higher than that in cycle 4. In the first two cycles, aluminum uptake was negligible in both MAP and Al-P sludge treatments; however, in cycles 3 and 4, there was significantly more Al in maize from sludge amended pots. Our results show that Al-P sludge was as effective as MAP in supplying enough P for biomass yield. We, therefore, conclude that Al-P sludge could be an alternative source of P, especially for growing maize as feedstock for bioenergy.

Corresponding author
Qiuyan Yuan, qiuyan.
yuan@umanitoba.ca

## INTRODUCTION

To sustain future economic growth, fertilizer usage will continue to increase due to the increasing population and corresponding food demand (*Schroder et al., 2011*). With several speculations of pure phosphate rock extinction, there may be a looming crisis in the nearest future (*Liu et al., 2020*). Improved agronomic practices and soil fertilization are expedient to alleviate phosphorus (P) pressured demands. With an over-reliance on mineral fertilizers, the agronomic sector faces a near global-P challenge and increasing environmental concerns as global P usage efficiency is reported to be about 25% on average (*MacDonald et al., 2011*; *Dhillon et al., 2017*; *Liu et al., 2020*; *Sattari et al., 2012*). In the face of this scarcity, with the agricultural sector accounting for about 90% of P demand (*Maghsoodi, Ghodszad & Lajayer, 2020*; *Childers, Corman & Edwards, 2011*), focus on P recovery from Al-P sludge should be prioritized.

P is abundantly found in wastewater sludge, and depending on the treatment method applied, it could be chemically or biologically bound (*Ojo & Ifelebuegu, 2018*). With the abundance of chemical sludge (aluminum phosphate) in Manitoba, Canada, aluminum-phosphate (Al-P) sludge is mostly landfilled. This disposal method is quite expensive (*Keeley, Jarvis & Judd, 2012*; *Babatunde & Zhao, 2007*). P recovery from chemical sludge is one of the ways to ensure a cleaner and more sustainable environment as well as ensure that all sludge P are put to beneficial use (*Tolofari et al., 2021*; *Alvarenga et al., 2016*; *Mazen, Faheed & Ahmed, 2010*).

As a widely studied crop, maize is mostly known for its feed generation for humans and animals. It is the third-ranking cereal and a very critical staple for the world's growing population (*Food and Agricultural Organization Faostat, 2017*; *Ranum, Pena-Rosas & Garcia-Casal, 2014*). In recent times, it is a widely sought-after feed for ethanol production. Maize generates more ethanol due to cellulose conversion technology (*Schwietzkey et al., 2009*). With an increasing demand for maize in bioethanol production, there will be intensifying competition and a surge in demand for all first users (man and animal). Maize cultivation and production should be intensified to secure its availability through a rising global demand scenario. *Rosegrant et al. (2009)* reports a shortage in current production compared to demand, which is most likely to double by 2050 in developing countries (*Shiferaw et al., 2011*). A significant challenge to achieving this surplus production is hinged on the reports of depleted P reserves (*Cordell, Drangert & White, 2009*), with fertilization cost seen as a major contributor making maize less affordable (*Omenda et al., 2021*; *Pereira et al., 2020*; *Metson et al., 2016*). P usage for maize is crucial at the initial growing season and can primarily affect yield and uptake (*Cadot et al., 2018*; *Lino et al., 2018*; *Grant et al., 2001*). Root development and distribution relative to the P in soil solution determine the extent of P uptake by maize (*Klamer et al., 2019*; *Mollier & Pellerin, 1999*). With a higher initial P concentration in soil solution, plants have a better initial root development and subsequent yield increase (*Dhillon et al., 2017*; *Wu et al., 2015*).

Therefore, reuse options for Al-P sludge are viable to enhance the reduction of GHG emissions; this can be achieved by the cropping of maize solely to produce ethanol as a
biofuel (*Tolofari et al., 2021*; *Maghsoodi, Ghodszad & Lajayer, 2020*). This would reduce cost competition against first users better the environment and prevent groundwater contamination (*Hoekman, Broch & Liu, 2018*; *Chiodo et al., 2018*).

Al-P sludge is rich in organic carbon and major nutrients such as N, P and K, which are beneficial to crops (*Kluczka et al., 2017*; *Kim et al., 2002*). The use of Al-P sludge for growing crops is a cost-effective way of recovering P as well as a means to improve overall soil properties that facilitate plant growth (*Kim et al., 2002*). This amendment can fill some gaps in the P industry and also foster growth in the agricultural industry. The risk of Al toxicity has been highlighted as a limitation to the use of Al-P sludge (*Kluczka et al., 2017*; *Kim et al., 2002*); however, creating the right balance in terms of soil conditions and application rate could allow for the safe use of Al-P sludge as a nutrient source for crop plants. The reuse of Al-P sludge as a P recovery option is a crucial step in addressing some of the challenges with the forecasted depletion of global P reserves and to reduce the dependence on inorganic fertilizers. Phosphorus availability in Al-P sludge could be dependent on the organic matter content and pH of the soil (*Chowdhury et al., 2021*; *Tolofari et al., 2021*).

The objective of this study was, therefore, to investigate the effectiveness of Al-P sludge as a source of P compared to a commercial fertilizer (monoammonium phosphate) at 9.7, 19.4, 29.1 and 38.8 mg P kg$^{-1}$ dry soil. We hypothesize that (i) P uptake by maize is greater in MAP-amended soil than in soil amended with Al-P sludge, (ii) maize biomass yield increases with P application rate and (iii) maize biomass yield decreases as the number of cropping cycles increases.

## MATERIALS AND METHODS

### Phosphorus sources

Dewatered sludge with 18% solids content was sourced from the Headingley wastewater treatment plant in Manitoba, Canada (49°52′25.4″N; 97°26′16.1″W), while monoammonium phosphate (MAP, a soluble inorganic commercial fertilizer containing 11% nitrogen and 52% phosphate) was obtained from a local distributor in Canada. Application of P to potted soil was based on the available P concentration in the Al-P sludge to enable comparison between sludge and MAP treatments. Since a large percentage of total P in the sludge is organic and unlikely to be readily available for plant uptake, applying P on the basis of bicarbonate-extractable P (Olsen-P; *Olsen, Cole & Watanabe, 1954*) content in Al-P sludge was more appropriate. Application of sludge based on available P (Olsen-P) should be carefully considered as a part of the non-labile P pool could mineralize during the cropping period. Thus, sludge application based solely on available P could lead to the addition of excess available P than intended (*Tolofari et al., 2021*). Chemical properties of the Headingly sludge have previously been reported (*Tolofari et al., 2021*) with a moisture content of 82% and total phosphorus (TP) concentration of 11,560 mg kg$^{-1}$.

**Table 1 Chemical properties of the soil.**

| Properties | Value |
| --- | --- |
| Nitrate (mg kg$^{-1}$) | 33.6 |
| Olsen P (mg kg$^{-1}$) | 3 |
| Potassium (mg kg$^{-1}$) | 46 |
| Organic C (g kg$^{-1}$) | 25 |
| Carbonates (CCE) (g kg$^{-1}$) | 12 |
| Soil pH | 8.2 |
| Cation exchangeable capacity cmol$_c$ kg$^{-1}$ | 25.1 |
| Sulfur (mg kg$^{-1}$) | 26.9 |
| Baron (mg kg$^{-1}$) | 2 |
| Zinc (mg kg$^{-1}$) | 0.44 |
| Iron (mg kg$^{-1}$) | 32.3 |
| Manganese (mg kg$^{-1}$) | 1.7 |
| Copper (mg kg$^{-1}$) | 0.2 |
| Magnesium (mg kg$^{-1}$) | 912 |
| Calcium (mg kg$^{-1}$) | 3,402 |
| Sodium (mg kg$^{-1}$) | 86 |
| Electrical conductivity (dS m$^{-1}$) | 0.25 |

## Soil

Topsoil (0–15 cm) from Roseisle, Manitoba, Canada (49°33.577′N; 98°24.824′W) was used for this study. The soil was a well-drained Entisol (dark-grey Gleyed Regosolic sand) with an initial pH of 8.2 and a Olsen P concentration of 3 mg kg$^{-1}$ (*Tolofari et al., 2021*).

Chemical properties of the soil have been previously published (*Tolofari et al., 2021*) and are summarized in Table 1.

## Experimental set-up

The investigation was conducted in a growth chamber where the temperature regime was maintained at 25/15 °C for day/night respectively, with a relative humidity of 60%, and a proportionate photoperiod of 16 h. The experimental design was completely randomized; the set-up was with a factorial treatment structure involving two P sources (MAP and Al-P sludge), four P application rates (9.7, 19.4, 27.1 and 38.8 mg P kg$^{-1}$ dry soil) and four cropping cycles. Sludge P application was based on Olsen P concentration. The sludge application rates were 12, 24, 36 and 48 g kg$^{-1}$ dry soil to achieve the above stated P application rates. All treatments, including the control, were replicated three times. On a dry weight basis, P application rates corresponded to 9.8, 19.7, 29.5 and 40 kg ha$^{-1}$ P. Typical agronomic rates for corn in Manitoba are within the range 20 to 25 kg ha$^{-1}$ P (*Manitoba Soil Fertility Advisory Committee, 1990*).

The soil was meticulously mixed and passed through a 4 mm sieve to remove stones and roots, after which it was air-dried and properly stored for analysis. Approximately 1.5 kg of air-dried soil sample was put into each 14.9 cm diameter by 16.5 cm height plastic pots. Reverse osmosis (RO) water and a full-strength nutrient solution without P (*Zvomuya,*

*Rosen & Gupta, 2006*) were added to all pots to bring the soil moisture content to field capacity. The pots were then stored in a growth room for 24 h prior to the commencement of planting.

Three maize seeds were planted into each pot; the seedlings were later thinned down to one 5 d after germination. The pots received two applications of essential macronutrient solutions (P excluded) in each growth cycle (*Zvomuya, Rosen & Gupta, 2006*). All pots were weighed and watered every other day to replace moisture lost *via* evapotranspiration. A total of four cropping (seeding to harvest) cycles were completed, with each cycle lasting 45 d in duration. Plants were harvested on day 45 by clipping at 2 cm height above the surface of the soil. The contents of each pot were emptied into a clean tray and thoroughly mixed. A 20-g soil subsample was collected at the end of every growth cycle for analysis. Plant roots were recovered, cut into small pieces, and mixed with the soil, which was re-potted for the next cropping cycle. The soil was stored in the growth room for 5 d before planting. Harvested biomass was weighed and oven-dried for 72 h at approximately 60 °C. Oven-dried biomass samples were later weighed, chopped into tiny pieces, and fine-ground in a mill grinder (<0.2 mm, Model 8000D; Spex Sample Prep, Metuchen, NJ, USA). At the completion of the fourth cycle, plant roots were retrieved, washed with RO water, dried, and ground for analysis.

## Laboratory analysis

Preliminary analysis before bioassay, the plant available (Olsen) P was analyzed for the soil using the ascorbic acid-molybdate method (*Murphy & Riley, 1962*) with a Skalar SAN++ segmented flow analyzer (Skalar Analytical B.V., Breda, Netherlands) after extraction with 0.5 M concentration of $NaHCO_3$ at pH 8.5 (*Olsen, Cole & Watanabe, 1954*); concentration of nitrate (using Cd reduction method; *Mulvaney et al., 1996*); pH of the soil and electrical conductivity (1:1 soil to water suspension); exchangeable K, Mg, Ca, and other cations (using atomic absorption spectroscopy after extraction of 5 g of soil with 33 mL of 1 M concentration $NH_4OAc$ at a pH of 7 (*Hendershot, Lalande & Duquette, 2008*)); concentration of aluminum (Al) in the soil was measured with an inductively coupled plasma optical spectrophotometer (ICP-OES, Thermo Electron ICAP 6300 Radial) after the digestion of the samples in block digester using sulfuric acid (99.7% $H_2SO_4$) and hydrogen peroxide (30% $H_2O_2$) (*Parkinson & Allen, 1975*); cation exchange capacity (sum of exchangeable Na, K, Ca and Mg) (*Hendershot, Lalande & Duquette, 2008*); concentration of soil organic matter (SOM) (using loss on ignition method (*Nelson & Sommers, 1996*)); and carbonates were analyzed using the pressure calcimeter method (*Loeppert & Suarez, 1996*), represented as calcium carbonate equivalent (CCE). Soil organic carbon (SOC) concentration was obtained by multiplying SOM by 0.58.

Total P (TP) and aluminum (Al) concentrations in the plant tissue were measured with an inductively coupled plasma optical spectrophotometer (ICP-OES, Thermo Electron ICAP 6300 Radial) after the digestion of the plant tissue samples in a block digester using sulfuric acid (99.7% $H_2SO_4$) and hydrogen peroxide (30% $H_2O_2$) (*Parkinson & Allen, 1975*). Concentration of Olsen-P in the soil was determined by the $NaHCO_3$ (99.7%) extraction method (*Olsen, Cole & Watanabe, 1954; Olsen & Sommers, 1982*) at pH 8.5. The

phosphorus concentration in the extract was determined colorimetrically using the ascorbic acid-molybdate method (*Murphy & Riley, 1962*).

## Statistical analysis

To determine treatment and interaction effects, analysis of variance (ANOVA) was carried out on P concentration in the soil, dry matter yield, P uptake and PRE data using PROC GLIMMIX for repeated measures (with cycle as the repeated measures factor) in SAS 9.4 (*SAS Institute, 2013*). Phosphorus recovery efficiency (PRE) was computed as ((P uptake in each amended pot – P uptake in control pot)/P application rate) for each harvest. Phosphorus source and P rate were modeled as fixed effects. Data for all variables were normally-distributed, except for Olsen P data which were analyzed as a lognormal distribution and PRE as a beta distribution. Treatment means were compared using the Tukey multiple comparison procedure at α = 0.05.

# RESULTS AND DISCUSSION

## Dry matter yield

We observed a significant effect of P source on maize DMY (Table 2). Averaged across all rates and cycles, pots amended with sludge had a significantly higher DMY than MAP amended treatments. *Ramphisa & Davenport (2020)* reported a different result as no significant effect of P source (MAP and organic manure) was observed on corn yield in their study. The authors attributed the lack of P source effect on maize yield to the short time (2 years) of the study. The same was true for the study on maize using alum sludge where there was no recorded significant DMY although this was attributed to the type of seed used (*Lin & Green, 1990*). The higher yield observed in our study for sludge amended soils could be due to the increase in inorganic P from sludge total P, as organic P can be transformed to inorganic P with time (*Shen et al., 2011*). Another reason could be that the sludge addition increased the organic matter content present in the soil coupled with the plant residue/root incorporated after each cycle. This could have increased microbial activities hence the overall increase of DMY (*Gatiboni et al., 2008*).

Regardless of the amendment applied, the rate effect on DMY varied with the cropping cycle (Table 2; Fig. 1). For cycle 1, as the rate increased from 9.7 to 19.4 mg P kg$^{-1}$ soil, we observed a significant increase in DMY. Between 19.4 and 29.1 mg P kg$^{-1}$ soil, there was no significant difference between DMY. However, a significant increase in DMY was identified between 29.1 and 38.8 mg P kg$^{-1}$ soil. For cycles 2 and 3, DMY was similar across all rates, no significant effect of rate on DMY. For cycle 4, DMY was similar for rates 9.7, 19.4 and 29.1 mg P kg$^{-1}$ soil. DMY was significantly higher at 38.8 mg P kg$^{-1}$ soil in contrast to the lowest rate, 9.7 mg P kg$^{-1}$ soil. However, the DMY at 38.8 mg P kg$^{-1}$ soil was similar to DMY at 19.4 and 29.1 mg P kg$^{-1}$ soil. Similar to what was observed in cycle 1, *Lino et al. (2018)* observed an increasing rate effect on corn yield. *Ortas & Islam (2018)* observed a non-linear increase in corn yield with respect to increasing P application rate. They attributed the increased yield to higher soil P availability due to mycorrhizal root interactions and mineralization of soil organic material (*Ortas & Islam, 2018*). For our study, a low-P soil was used; however, since some P was supplied in the first cycle, it could

**Table 2 DMY, phosphorus uptake, PRE, aluminum uptake, pH, and EC as affected by sludge and MAP application.**

| Effect | DMY g kg⁻¹ | P uptake mg P kg⁻¹ | PRE % | Al uptake mg Al kg⁻¹ | pH | Olsen P mg P kg⁻¹ | Soil TP mg P kg⁻¹ |
|---|---|---|---|---|---|---|---|
| **Cycle (C)** | | | | | | | |
| 1 | 10.8 | 11.1a | 23.6 | 0.1b | 7.9a | 13.4ab | 563a |
| 2 | 3.5 | 4.8c | 11.4 | 0.0b | 7.0c | 22.4a | 510b |
| 3 | 3.7 | 7.1b | 17.0 | 4.9b | 7.1c | 12.4ab | 405c |
| 4 | 2.5 | 4.0c | 8.2 | 13.3a | 7.6b | 10.0b | 424c |
| **P-source (P)** | | | | | | | |
| Control | 1.6c | 2.6c | | 2.2b | 8.1a | 10.2 | 456b |
| Sludge | 6.05a | 9.7a | 30 | 6.0a | 7.6b | 14.1 | 510a |
| MAP | 4.3b | 3.8b | 3 | 3.1b | 7.2c | 13.3 | 440b |
| **Rate (R, mg P kg⁻¹)** | | | | | | | |
| 0 | 1.68 | 2.6d | | 2.2 | 8.05a | 10.2 | 456 |
| 9.7 | 4.4 | 4.8c | 14.5 | 4.7 | 7.3c | 11.1 | 446 |
| 19.4 | 5.2 | 6.5b | 16.4 | 5.4 | 7.3c | 12.0 | 486 |
| 29.1 | 5.1 | 7.3b | 13.6 | 4.3 | 7.4bc | 14.6 | 478 |
| 38.8 | 5.7 | 8.4a | 13.9 | 3.9 | 7.5b | 18.0 | 491 |
| | | *P*-value | | | | | |
| C | <0.0001 | <0.0001 | <0.0001 | <0.0001 | <0.0001 | <0.0001 | <0.0001 |
| P | <0.0001 | <0.0001 | <0.0001 | <0.0001 | <0.0001 | 0.57 | <0.0001 |
| R | <0.0001 | <0.0001 | 0.68 | 0.59 | 0.007 | 0.007 | 0.06 |
| P × C | 0.38 | <0.0001 | <0.0001 | 0.001 | <0.0001 | 0.49 | 0.14 |
| P × R | 0.30 | <0.0001 | 0.01 | 0.18 | 0.89 | 0.80 | 0.13 |
| R × C | 0.04 | 0.01 | 0.28 | 0.88 | 0.09 | 0.91 | 0.10 |
| P × R × C | 0.62 | 0.002 | 0.35 | 0.20 | 0.97 | 0.86 | 0.13 |

**Note:**
Letters a, b, c, d indicate the significance of the mean components. The same letters show that the means are not significantly different. Means followed by different letters within a column are significantly different at $P \leq 0.05$.

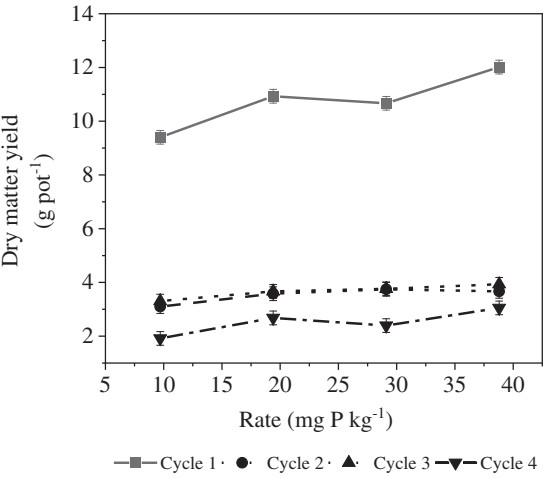

**Figure 1 Effect of the cropping cycle × rate interaction on maize dry matter yield.**

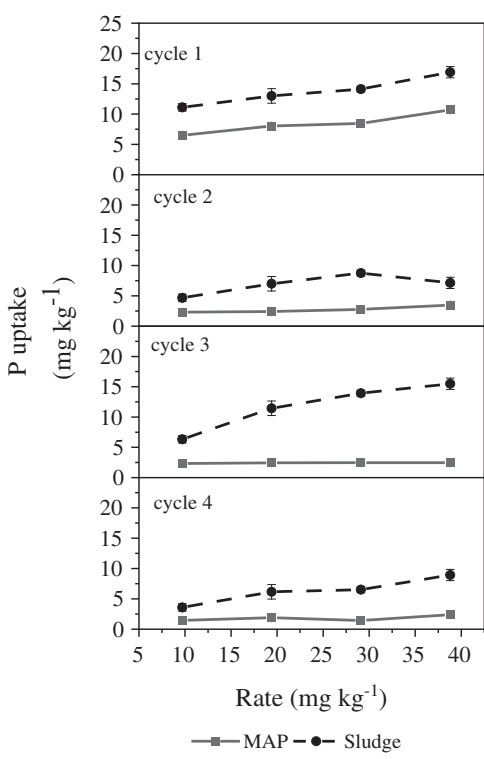

**Figure 2 Effect of rate × cycle × P source interaction on P-uptake by maize.**

be that this increased the available P pool thus removing any rate effect in cycles 2 and 3 (*Lopes et al., 2007*). The increase in DMY for the highest rate in cycle 4 could be because of an interaction effect between soil processes and mineralization of incorporated roots (*Ortas & Islam, 2018*; *Martinazzo et al., 2007*; *Santos, Gatiboni & Kaminski, 2008*).

Sludge, which is rich in organic matter and major nutrients, can improve soil physical and chemical properties which are important for plant growth (*Rodríguez-Berbel et al., 2022*), and therefore yield. The increase in DMY must have been improved by soil properties such as CEC, organic carbon, carbonates etc. The addition of Al-P sludge to the soil resulted in an increase in SOM, which then improved the fertility and properties of the soil (*Chowdhury et al., 2021*). For example, *Chowdhury et al. (2021)* explained that CEC above 10 $cmol_c$ $kg^{-1}$ soil would improve plant growth as well increase biomass yield. This would also improve uptake of micronutrients and prevent deficiency. The presence of organic carbon improves the availability of nutrients by the continuous decomposition and breaking down of bound nutrients into readily usable forms which would improve crop growth and invariably the dry matter yield of the plant. Organic matter decomposition facilitates nutrient availability and uptake by roots, thus increasing DMY (*Chowdhury et al., 2021*; *King et al., 2020*).

## Phosphorus uptake

Phosphorus uptake showed a significant amendment by rate by cycle interaction ($P < 0.002$) (Table 2, Fig. 2). For cycle 1, as the rate increased from 9.7 to 38.8 mg P $kg^{-1}$

soil, we observed a significant increment in P uptake by maize in both MAP and sludge amended soils. Interestingly, P uptake by maize was significantly higher in sludge amended soils than in MAP at all application rates. For both P sources, the highest P uptake by maize occurred at the highest application rate. This report collaborates what *Pizzeghello et al. (2019)* and *Yossif & Gezgin (2019)* observed. The authors reported an increased P uptake as the application rate of MAP/single superphosphate and MAP/DAP increased (*Pizzeghello et al., 2019*; *Yossif & Gezgin, 2019*). *Silva et al. (2012)* also reported that the highest P uptake was at the highest P application rate.

For cycle 2, P uptake in sludge amended pots was significantly higher at each rate compared to MAP amended pots, this observation is similar to cycle 1. As the rate increased for MAP, there was no significant increase in P uptake. For sludge amended soils, there was no significant difference in P uptake at these application rates (9.7, 19.4, and 38.8 mg P kg$^{-1}$ soil). The highest P uptake from sludge amended soils occurred at rate of 29.1 mg P kg$^{-1}$ soil and was significantly higher than P uptake in pots amended with rate 9.7 mg P kg$^{-1}$ soil. However, in cycle 3, there was a significant increase in P uptake between rate 9.7 and 38.8 mg P kg$^{-1}$ soil for sludge amended soils. As the application rate increased, no significant increase in P uptake from MAP amended soils was observed for cycle 3. For cycle 4, similar to what we observed in cycle 3, there was no significant increase in P uptake with increasing P rate in MAP amended soils. However, P uptake increased between rate 9.7 and 29.1 mg P kg$^{-1}$ soil. P uptake in pots amended at rate 29.1 mg P kg$^{-1}$ soil was not significantly different to P uptake at 38.8 mg P kg$^{-1}$ soil, for sludge amended soils.

For MAP and sludge amended soils, the highest P uptake was observed for cycle 1. This was not surprising for MAP as it is an inorganic source of P, and all its P is in the readily available form. Readily available P in MAP may have been precipitated by calcium and magnesium in the soil, thus reducing its availability to plants (*Tolofari et al., 2021*). Sludge, being an organic source of nutrients has both organic and inorganic pools of P, with the organic pool being a long-term source of available P with a gradual release pattern for other growth cycles (*Ahmed, Fawy & Abdel-Hady, 2010*; *Kiehl, 2008*). This provides an explanation of the differing trends we observed in the different cycles as the organic P becomes more available. In cycles 1, 2 and 3, at each P rate, P uptake in sludge amended soils was significantly greater than MAP amended pots. However, in cycle 4, at 9.7 mg P kg$^{-1}$ soil amendment application rate, there was no observed significant difference in P uptake between sludge and MAP amended pots. At other rates, the trend was similar to those observed in previous cycles where P uptake for sludge treatments was higher. The similar P uptake in cycle 4 for both MAP and sludge at the lowest rate (9.7 mg kg$^{-1}$ soil) could be explained by the complete mineralization of all the organic P in the sludge as a consequence of the lower P supplied at this rate (9.7 mg kg$^{-1}$ soil) (*Santos, Gatiboni & Kaminski, 2008*; *Martinazzo et al., 2007*). It is also possible that newly developing roots do not have good contact with soil solution, hence reducing P uptake from the pot (*Rehim et al., 2016*).

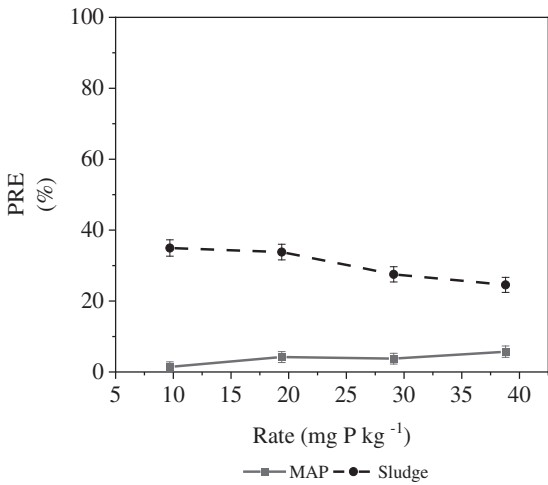

**Figure 3  P source × rate interaction on PRE.**     

## Phosphorus recovery efficiency

There was a significant P source × cycle interaction and P source × rate interaction. For sludge, there was no significant effect of rate on PRE, while for MAP, there was a numerical increase in PRE with increasing P application rate (Fig. 3). PRE for MAP amended soils was significantly higher at 38.8 mg kg$^{-1}$ soil compared to 9.7 mg kg$^{-1}$ soil. Similar to our Al-P sludge treatment, *Sá et al. (2017)* reported no significant change in PRE across all rates of P application for both organic and inorganic P sources (50–400 mg P kg$^{-1}$ soil). The authors also observed no significant difference between PRE of MAP and OMF (*Sá et al., 2017*). This was not the case in our study as at every application rate, sludge had a significantly greater PRE than MAP. Although, *Sá et al. (2017)* expected differences due to the different formulations of the fertilizers and the uniqueness of the contact surfaces, which could affect P availability, they observed none. They however concluded that the recovery efficiency with the high DMY makes poultry litter a competitive source of P. In our study, the higher PRE for sludge compared to MAP might be due to an increase in the bioavailability of the P in Al-P sludge when in contact with the calcareous soil (*von Wandruszka, 2006*). Since P in MAP is in the readily available form, its P bioavailability in calcareous soils could be limited due to the formation of less soluble forms through its interaction with Ca and Mg hence leading to precipitation (*Weeks & Hettiarachchi, 2019*; *Naeem, Muhammad & Waqar, 2013*; *Syers, Johnston & Curtin, 2008*).

*Tolofari et al. (2021)* reported that PRE for sludge amended pots had shown the highest recovery efficiency at the lowest P application rate (9.7 mg kg$^{-1}$). At the other P rates in their study, no significant difference in PRE at was observed (*Tolofari et al., 2021*); this is similar to our observation. *Yossif & Gezgin (2019)*, who worked on maize, observed that P recovery for calcareous soils is low, with the highest recovery observed at 30 mg kg$^{-1}$ P$_2$O$_5$ (13.77%) and the least recovery at 90 mg kg$^{-1}$ P$_2$O$_5$ (9.36%). Low PRE observed at higher P application rates could mean that maize used just a small fraction of the applied rate (*Rehim et al., 2012*).

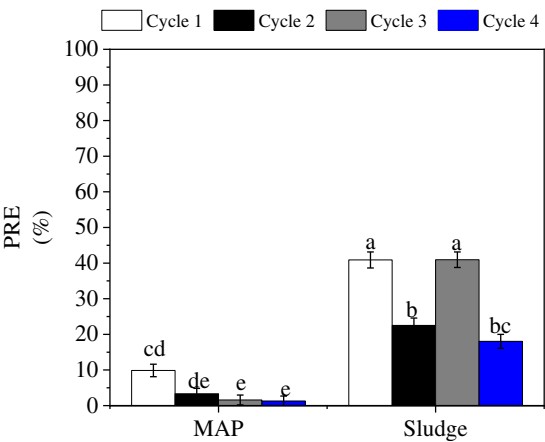

**Figure 4 P source × cycle interaction on PRE.** Letters a, b, c, d, e indicate significance on the mean components; bars with the same letters are not significantly different.

There was also a significant P source by cycle interaction (Fig. 4). In all cycles, PRE of sludge was significantly higher than MAP. For MAP, PRE decreased with cycle. This is mostly due to the possible fixation of available P due to the calcareous nature of the soil, and this is also aggravated by the one-time P source addition which was prior to the first cycle. In contrast, no increasing or decreasing trend was observed for sludge amended soils in all four cycles. The varying PRE for sludge is similar to the trend we observed for P uptake and could be attributed to the slow release of available P from the organic P pool in sludge (*Iyamuremye, Dick & Baham, 1996*). The lowest P uptake was observed in cycle 4, which explains the PRE in cycle 4. *Tolofari et al. (2021)* observed an increase in PRE with cycle which was dissimilar to our study, as we observed decreasing PRE with cycle (for MAP) and no trend for sludge. This difference could be due to the effect of the plant/ organic residue which was re-introduced into the pots, and which must have increased biological activities in the soil (*Smith et al., 1993*; *Nziguheba et al., 1998*).

### Aluminum uptake

There was a significant P source × cycle interaction on aluminum uptake, indicating that aluminum uptake in each cycle was dependent on the P source (Fig. 5). In cycle 1 and 2, Al uptake by MAP and sludge was similar (Fig. 5). It is important to state that the Al uptake was very close to zero in cycle 1 and 2 (Table 2). In cycles 3 and 4, there was significantly more Al uptake in sludge amended soils compared to MAP amended soils, with the uptake in cycle 4, higher than uptake in cycle 3. This suggests an effect of time as well as P source on Al release from soil. Unfortunately, Al in soil was not quantified at the start of the study. The Al in sludge could have contributed to the increased Al uptake from sludge amended pots. The mineralization of organic matter may have caused an increase in the uptake of Al by the plants (*Meriño-Gergichevich et al., 2010*; *Hede, Skovmand & López-Cesati, 2001*).

Maize is reportedly a very sensitive crop to Al and high levels of Al limits biomass production and P uptake, even at very low Al application rates and short exposure intervals (*Giannakoula et al., 2008*; *Yossif & Gezgin, 2019*; *Tandzi et al., 2018*). The effect of Al in

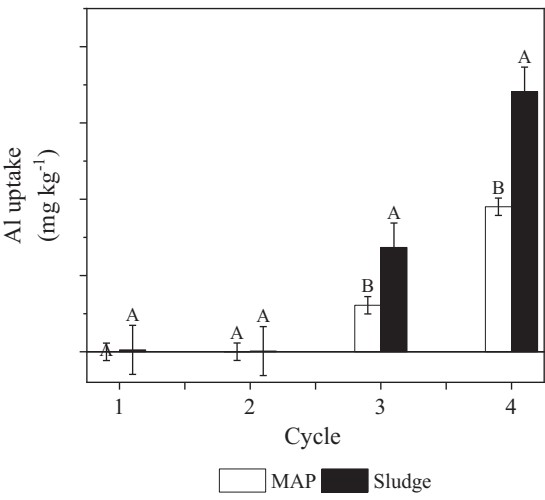

**Figure 5 Effect of P source × cycle on aluminum uptake.** Letters A and B indicate significance on the mean components. The same letter error bars are not significantly different.

alkaline and acidic soils negatively impact root growth, however acidic soils have much more severe impact on root reduction (*Stass et al., 2006*; *Lidon & Barreiro, 2002*). In this work, the effect of Al was not noticeable as Al uptake did not occur in the first two cycles. This could be due to the pH of the soil which was at 8.2 at the start of the experiment. *Mimmo et al. (2009)* explained that prominent levels of Al in soil may affect nutrient uptake; however, their experiment on beans showed that P uptake was not significantly limited by Al in soil even at a pH of 4.50. Although the species of Al in the soil were not assessed, Al can be organically bound, exchangeable or in crystalline form (*Kluczka et al., 2017*). The Al species expected at the pH in our study are $Al(OH)_3$, $Al(OH)^{2+}$ and $Al(OH)_2^+$ (*Meriño-Gergichevich et al., 2010*). Understanding the speciation of Al in the soil explains a lot about their interaction. In this study however, Al uptake occurred even in the control which could be due to the solubility of Al in the soil. *Kluczka et al. (2017)* explained that increase in Al around neutral pH is mostly due to exchangeable Al in the soil and Al bound to organic matter.

The duration of Al species incubation in the soil can also affect its bioavailability (*Iqbal, 2012*).

## pH

There was a significant P source × cycle interaction for soil pH (Fig. 6, Table 2). In cycles 1 and 2, the pH did not differ significantly between sludge and MAP. It is noteworthy that, in cycle 1, the pH of both MAP and sludge amended soils were in the alkaline range, while in cycle 2, the pH of both P source treated soils were near neutral indicating a reduction in pH in cycle 2 (Fig. 6). In cycle 3 however, the pH for MAP amended soils was in the acidic range (6.3) and was significantly lower than pH for sludge amended soils (7.8). This reduced pH in MAP amended pots in the third cycle may be due to the excretion of organic acids by the roots in the soil. Since available P may have been diminished, organic acid

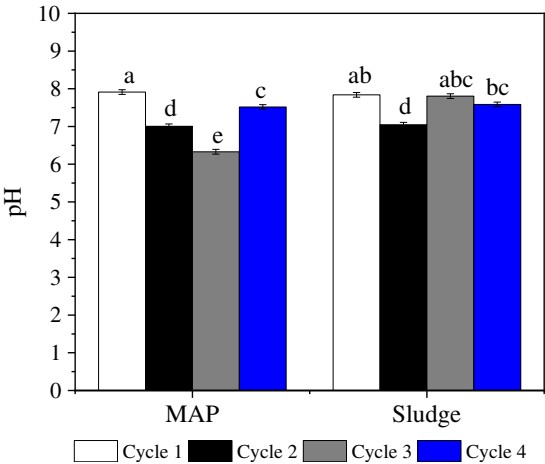

**Figure 6 Rate × cycle interaction on soil pH.** Letters a, b, c, d, e indicate significance on mean components, bars with same letters are not significantly different.

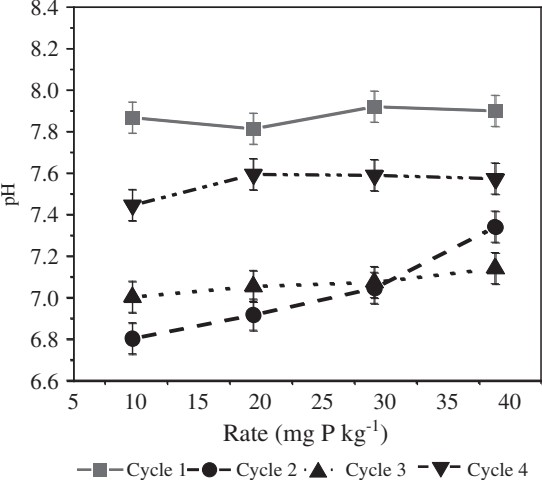

**Figure 7 P source × cycle interaction on soil pH.**

excreted by roots may be able to solubilize P from metal P salts and make it available for utilization (*Hinako et al., 2021*). In cycle 4, there was no significant difference between pH of MAP and sludge amended soils (Table 2). *Thé et al. (2006)* reported a reduction in soil pH of about 0.23 units upon continuous maize cropping using organic and inorganic fertilizer in acidic soil of pH 4.8. Similar to *Thé et al. (2006)*, we observed a decreasing trend for MAP until cycle 3, however there was an increase in pH in cycle 4. This alkalinity reduction could mostly be due to the buffering capacity of the soil (*Keramati, Hoodaji & Kalbasi, 2010*).

Generally, at the soil pH in cycles 3 and 4, we would not expect high Al exchange in the soil. The increased uptake of Al in the 3[rd] and 4[th] cycles for alum-P pots could be part of the effect of incubation duration and an effect of organic matter mineralization due to the incorporation of roots into the pots after each cycle (*Grzyb, Agnieszka & Niewiadomska,*

*2020*; *Hadas et al., 2004*). *Schwesig, Kalbitz & Matzner (2003)* highlighted that the mineralization of organic matter in the soil would result in increased concentrations of free $Al^{3+}$ in soil solution.

There was also a significant rate effect on soil pH (Fig. 7). Across all cycles and both P sources, there was a numerical increase in pH with amendment rate. Soil pH was not significantly different between rates 9.7 and 29.1 mg $kg^{-1}$, however, pH in soil amendment with the highest P rate (38.8 mg $kg^{-1}$) was significantly higher than pH in pots amended at P rates of 9.7 and 19.4 mg $kg^{-1}$. Contrary to our finding, *Arvas, Çelebi & Yilmaz (2011)* reported that the lowest pH values were identified in pots of the highest application rate of sewage sludge; this they attributed to the effect of root biomass in the soil. *Ahmad et al. (2019)* observed the highest values in units with the highest DAP application rates while single superphosphate showed a slight decrease in pH.

### Olsen P and TP

There was a significant effect of P rate on soil Olsen P (Table 2). As expected, there was a numerical increase in soil Olsen P as the amendment rate increased. At the lower P application rates, there was no significant difference in soil Olsen P between P sources. The highest Olsen P recorded in our study was at the highest P application rate. *Wen et al. (2016)* and *Zhan et al. (2015)* indicated that the increase in Olsen-P at higher rates was as a result of increased decomposition of organic matter in the soil. *Rehim et al. (2016)* also found a similar increase in Olsen-P with respect to increasing rate of P application while lower Olsen-P values could be attributed to efficient P usage by the crops.

There was also a significant cycle effect on soil Olsen P. The highest soil Olsen P was observed in cycle 2, and this was significantly greater than Olsen P in other cycles. Overall decrease in Olsen-P could have resulted in declining PRE trend observed for MAP.

For soil total P, there was a significant cycle and P source effect. Total P decreased with cycle until cycle 3, and no significant change was observed in cycle 4. TP trend was cycle 1 > cycle 2 > cycle 3 = cycle 4. This was as expected as it clearly shows that accumulated soil P was being utilized for subsequent crop growth (*Zhang et al., 2020*). TP measured in sludge amended pots was significantly greater than in MAP amended pots. This was not surprising as sludge properties showed that it contained a large amount of total P (*Tolofari et al., 2021*).

## CONCLUSIONS

Our results demonstrate the effectiveness of P uptake from Al-P sludge using maize as a test crop in a growth room. Maize yield significantly increased with the application of Al-P sludge compared to MAP; however, there was a general decrease in biomass yield as the number of cropping cycles increased. PRE significantly increased with Al-P sludge application relative to conventional fertilizer. In addition, as the application rate increased, there was no significant difference in PRE for the different application rates of Al-P sludge. Generally, as the cropping cycle increased, P uptake decreased alongside TP and soil Olsen-P concentrations. Also in most cases, at all cycle and in all rates, P uptake from Al-P sludge amended soil differed significantly from MAP amended soil.

Overall, Al-P sludge could be an alternative source of P in a world with an ever-increasing P demand, thus reducing the demand for inorganic fertilizers such as MAP. This can promote the dedicated use of Al-P sludge to produce feedstock for biofuel, leading to diversified P usage. Based on the sludge application rates in this study, the lowest application rate of 9.7 mg P kg$^{-1}$ soil showed adequate PRE that is comparable to higher rates and hence may be the most suitable for future use to prevent overapplication. Although the one-time application of Al-P sludge is beneficial to subsequent crops, we recommend that care should be taken in the use of Al-P sludge as application based on available P may lead to P accumulation and pose a potential risk to the environment.

## ACKNOWLEDGEMENTS
We thank Rob Ellis and Mulikat Bamike for their technical support.

### Funding
The Niger Delta Development Commission (NDDC) provided Ayaobu Tolofari's graduate scholarship. The funders had no role in study design, data collection and analysis, decision to publish, or preparation of the manuscript.

### Grant Disclosures
The following grant information was disclosed by the authors:
The Niger Delta Development Commission (NDDC).

### Competing Interests
The authors declare that they have no competing interests.

### Author Contributions
- Ayaobu Tolofari performed the experiments, analyzed the data, prepared figures and/or tables, authored or reviewed drafts of the article, and approved the final draft.
- Theresa Adesanya analyzed the data, prepared figures and/or tables, and approved the final draft.
- Francis Zvomuya conceived and designed the experiments, analyzed the data, authored or reviewed drafts of the article, and approved the final draft.
- Qiuyan Yuan conceived and designed the experiments, analyzed the data, authored or reviewed drafts of the article, and approved the final draft.

### Data Availability
The raw measurements are available as a Supplemental File.

### Supplemental Information
Supplemental information for this article can be found online at http://dx.doi.org/10.7717/peerj.13885#supplemental-information.

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
