# Peer review of "Aluminum phosphate sludge as a phosphorus source for maize production under low soil phosphorus availability"

_PeerJ, doi:10.7717/peerj.13885_

## Round 0.1 · original submission · Major Revisions

Our reviewers have suggested that your manuscript requires revision before it can be further considered for publication. I invite you to revise your manuscript by giving due consideration to comments and suggestions from reviewers. I also suggest you convert line charts into bar charts and think of making multiple figures. Adding results from statistical analysis in figures will be more appropriate.

Reviewer 1 ·

Basic reporting

no comment

Experimental design

no comment

Validity of the findings

no comment

Additional comments

Dear Editor,
The submitted manuscript entitled ‘Aluminum Phosphate Sludge as a Phosphorus Source for Maize Production in a Low- Phosphorus Soil’ aimed to study Al-P sludge as a source of P compared to a commercial fertilizer (monoammonium phosphate, MAP). The article seems to have been elaborated carefully and my conclusion is that these data deserve publication in PeerJ. Therefore, I recommend reconsidering the paper after minor revision.

See specific comments:
Introduction: an in-depth explanation for using Al-P sludge as a P source is crucial in the introduction section.
Please, provide details on Al species in the sludge.

L78: “improve the ozone”. Please clarify the use of this term.

Material and methods
Authors should insert the corresponding rates of sludge applied (in kg ha-1) at each dose of P.
L150-157: and Al in the soil???

Results and Discussion
Dry Matter Yield: improve the discussion of the secondary effects of Alum-P sludge on dry matter yield (Other nutrients, organic matter, CEC, etc.)

Aluminum Uptake: knowing the chemical species of Al in the soil is crucial for a correct interpretation of results.

Improve discussion on the relationship between pH and exchangeable Al in soil. In this value of pH would not expect to find a high concentration of exchangeable Al in the soil (3 and 4 cycles).
Conclusions:
Describe a little more about the doses of sludge applied in the present study. Is it viable from an economic point of view??

·

Basic reporting

The work discusses the effects of an aluminium sludge in providing P for corn in a growth chamber over 4 cycles of 45 days each in comparison to the conventional source MAP. I congratulate the authors of this work. Although their findings are interesting and worthy of publication, I would like to suggest a few changes to further enhance the quality of the MS.
First, I would consider changing the name of the article from “Aluminum phosphate sludge as a phosphorus source for maize production in a low phosphorus soil” to “Aluminum phosphate sludge as a phosphorus source for maize production under low soil phosphorus availability”, because the total soil P is not that low when compared to, for example, tropical soils, and it is definitely not low when compared to the crops total demand.
Please explain in the introduction and with suitable references, why the use of sludge could be beneficial not only environmentally, but also in agronomical terms. Why should sludge provide more P to the plants then readily-available fully water-soluble phosphate sources?
L16. I would change the term “in response to” to “as a consequence of”.
L19. Please state the maize species (Zea mays L.) in the first appearance in the text rather than in the second.
L41-42. “Fertilizer usage will continue to grow…” needs a reference.
L72. “…higher initial P in the solution” – please define “soil solution”; “we get a better initial root development” – it is the plants who have a better root development. I would change to “plants have a better initial root development…”
L82-84. Hypothesis should always be written in the present tense. e.g. “… (i) P uptake by … would be greater than Al-P sludge” should be “P uptake by … is greater than Al-P sludge”
(ii) “Biomass yield will increase with ...” should be “Biomass yield increases with…”
Also, it is unclear what you mean by “decrease with cropping cycle”. Please explain.
L89. The phrase “…fertilizer mostly used in Canada” conveys the idea that Canada is one of the only places where MAP is used, which is not true. Please change to “the most used fertilizer in Canada” or something similar.
L93. You state that the bicarbonate-extraction is the Olsen P. Please reference the method in the same line, e.g. “Available P was extracted by the Olsen P method as described by Olsen et al., 1954. Also, stick to the term Olsen P from then on, do not switch between bicarbonate-extraction and Olsen P (as you do, for example, in L103 and 104).
L118. “A 1.5kg of soil” is missing something. “A 1.5kg sample” perhaps?
L123. “The chemical properties…published in Tolofari et al.” - It is crucial that the soil chemical properties of the soil you used is shown in this publication. The papers should be self-contained and sufficient for its own understanding. Researchers cannot have to go and dig on other works to have information that is relevant for this piece of work. Please adapt.
L 131, 134, 135. You mention P application rates in P and later the recommendations in P2O5. Please stick to P and change these values accordingly to make it easier to compare. The results and discussion section is good in that regard. Additionally, I recommend reading the letter “The Pervasive use of P2O5, K2O, CaO, MgO, and basic cations, none of which exist in soil” by Lambers and Barrow (2020). https://doi.org/10.1007/s00374-020-01486-5.
L157. Here you mention again the method by Murphy and Riley, but now with a different name. Please use the same you used in L105.
Overall, the Materials and Methods section is very well written and explained.
L256. I think the term precipitation is more suitable then fixation. Please check.
L305-314. Could it be that the pH reduction under MAP in the third cycle was due to plant exudation of organic acids in the intent to mine for P, since MAP is most likely gone in the third cycle specially in the lower doses?
L.318. “At the lower rates…no significant differences in Olsen P” – between what? Sources? Control?
L 332. “…a good amount” could be “high amounts”, since “good” is very arbitrary.
Overall, the language used in the MS is very clear, concise and correct. Congratulations.
The figures look good and clear, albeit a little bit small. The results are well grounded in literature and references are suitable.

Experimental design

Review the hypothesis (as suggested above) and make sure that the knowledge gap is clearly identified. State the research question clearly and why it is relevant.
Overall the methods used are clear and were well described. The experimental setup is scientifically correct and can be reproduced elsewhere. The technical and ethical standards of the research suit the requirements.

Validity of the findings

The research work was done carefully and the results are well explained and discussed. The dataset provided is robust and controlled. The statistical analyses performed are also coherent with the results obtained.
The discussion in L225-239 is not very conclusive. Please give an explanation as to why the sludge performed as well as MAP in cycles 1, 2 and 3 (even when it has less available forms of P) and no differences were seen between them in cycle 4, even when sludge contains organic P that should mineralize with time. Is this maybe a process that happened until the third cycle and was then depleted completely? How does that correlate with the P uptake and Olsen P values in this cycle?
Conclusions are concise and well grounded, but should be linked to the initial research questions when provided.

Additional comments

Some statements are not strictly wrong, but rather a personal preference. I, for instance, would not go for the phrase in L49. “the core for mankind is to focus on P recovery”. In my opinion, it is too bold and may not represent the reality. I suggest you reconsider, but again, it is a personal preference.
L. 63. “Achieving the right level of maize production to fortify against shortages and meet the world’s rising demand, one must expand the cultivation of maize.” This phrase also reads a bit strange to me. What is “the right level of maize production” and “one must expand…”. I would, for example, change to: “Maize cultivation and production should be intensified to secure its availability through a rising global demand scenario.
The word “expand” can also portray the meaning of an increase in area, which is not always the best way to go about crop production.

Reviewer 3 ·

Basic reporting

The formulated research objective should include the soil properties to be tested.
In Table 1, please complete the homogeneous groups for all experimental factors.
EC is shown in Table 1, but not discussed in the manuscript.
Figure 2 has been labeled as Figure 1, so there are two Figure 1s.
There are no references to Figures 2-7 in the text.

Experimental design

In the Materials and Methods chapter, in the Soil section, please leave only the data characterizing the soil material, but move the information on methods for determining individual soil properties to the Laboratory analysis section.
Please also move the information in lines 117-123 ("The soil...planting") to the Experiment Setup section.
Line 166: ‘correlation’? I suppose you meant ‘interaction’ between experimental factors.

Validity of the findings

Please add a conclusion on the soil properties tested.

Additional comments

Line 83: Put hypotheses (i) and (ii) together in one sentence.
Line 89: (MAP; 11-52-0) – these numbers must be explained
Line 98: TP - Explain the abbreviation when first using it.
Line 102: Check the language: “…and had a with an initial pH…”

---

## Round 0.2 · Minor Revisions

Our reviewers and I have looked at your revised manuscript. Your manuscript is improved during revision; however, there are some minor changes to be made further as suggested by Reviewer 3. I invite you to revise your manuscript accordingly before further consideration for publication.

Reviewer 1 ·

Basic reporting

No comment.

Experimental design

No comment.

Validity of the findings

No comment.

Additional comments

The article has been revised accordingly and deserve to be published in the present form.

·

Basic reporting

The authors did an excellent job in taking the comments provided by the reviewers and substantially improved the quality of the manuscript. Congratulations on putting together this interesting work. I have no further consideration for possible publication.

Experimental design

No additional comments

Validity of the findings

No additional comments

Additional comments

No additional comments

Reviewer 3 ·

Basic reporting

No comment

Experimental design

No comment

Validity of the findings

No comment

Additional comments

All my previous suggestions have been addressed. I only have the following minor comments remaining:
Line 104: “11-52-0” – this is unnecessary now
Table 1: I suggest changing “Roseisle” to “Value”

---

## Round 0.3 · accepted · Accept

I am pleased to inform you that your revised manuscript has been accepted for publication.